# One's Heaven Can Be Another's Hell: A Mixed Analysis of Portuguese Nationalist Fanpages

**Branco Di Fátima * and José Ricardo Carvalheiro ***

LabCom, University of Beira Interior (UBI), 6201-001 Covilhã, Portugal
* Correspondence: brancodifatima@gmail.com (B.D.F.); jose.carvalheiro@labcom.ubi.pt (J.R.C.)

**Abstract:** This paper analyzes the processes of racialization in Portuguese right-wing political movements through two prominent nationalist fanpages. It employs a mixed-methods approach that includes both quantitative and qualitative official data. The sample covers 72 months, from January 2017 to December 2020, encompassing a total of 3670 posts on Facebook. The main findings reveal that the fanpages utilize different discursive strategies, sometimes focusing on publishing static images and other times on sharing news links. From these publications, the fanpages garnered more than 1.4 million interactions, demonstrating consistent growth in their follower bases over the years. Emotional responses played a significant role in the interactions, particularly with Love, Sad, and Angry reactions standing out. The results also reveal that Portuguese nationalism maintains a dual ideology concerning race: ethno-exclusivism and ethno-pluralism. This observation affirms the dual nature of nationalist fanpages, where narrative elements converge and diverge based on the intended goal. Thus, individuals from Africa and Afro-descendants can be portrayed as both national heroes and social scum.

**Keywords:** nationalism; social media; far-right; decoloniality; black people; Portugal

## 1. Introduction

This paper explores the process of racialization in Portuguese right-wing political movements, focusing on two prominent nationalist Facebook fanpages. *Nova Portugalidade* (NP), established in February 2016, aims to reinforce the mythical idea of an imperial and multicultural country. However, this fanpage does not critically approach the processes of colonization in Africa. On the other hand, *Associação Portugueses Primeiro* (APP), founded in September 2017, is a cultural strand associated with the far-right party *Ergue-te*—a new designation of the National Renewal Party (PNR) since 2020. This fanpage strongly opposes immigration flows from former colonies and their descendants. Together, these online political movements generate millions of interactions annually, participate in events beyond the digital sphere, and even influence traditional media.

The analysis focuses on the online behavior of fanpage administrators and their followers, examining interaction and engagement rates. The objective is to answer the following question: what are the frequency and characteristics of racial allusions found within Portuguese nationalist movements on Facebook? For this, the research employs a mixed-methods approach, integrating both quantitative and qualitative techniques across three procedural stages: data extraction, mining, and visualization. Data were collected using *CrowdTangle* tools, covering a diverse dataset of 3670 posts spanning 72 months. This dataset includes details such as post date and time, URL addresses, content categories, message text, and follower reactions. For the qualitative analysis, the research chose a sub-sample of posts with the highest number of interactions and applied the concept of framing to emphasize their discursive nature.

The main results indicate that fanpages employ diverse discursive strategies, alternating between the emphasis on static image posts and the sharing of news links. These

posts collectively generated over 1.4 million interactions on Facebook, illustrating a steady expansion of their follower bases over time. Notably, emotional reactions comprised around 12.5% of these interactions, with expressions of Love, Sadness, and Anger being particularly prominent. These statistics underscore the dual nature of nationalist fanpages, where narrative elements converge and diverge based on the intended objective. Thus, individuals from former African colonies and their Portuguese descendants can be portrayed as both national heroes and social scum.

## 2. Race in Far-Right Nationalist Movements in Portugal

Defining nationalism as a political movement, it seeks to "attain and sustain the autonomy, unity, and identity of a population" (Smith 1997, p. 97). However, nationalism can also be regarded as an ideology with the goal of delineating the boundaries of that identity. In the 20th century, there were at least two strands of right-wing Portuguese nationalism that placed the issue of race at the core of their discourse. Both did so within the context of colonialism, albeit at different times and with distinct rationales, yet sharing points of intersection. The first strand, which emerged during the First Republic, is known as Lusitanian Integralism. The second, a response to the decline of the Portuguese empire from the 1950s to the 1970s, is referred to as Luso-Tropicalism.

Established in 1914 as a reaction against liberalism and republicanism, Lusitanian Integralism embraced anti-individualism and social hierarchization. This movement amalgamated aspects of 19th-century counter-revolutionary reactionism with contemporary fascist elements. Its primary preceptor, Antonio Sardinha, developed a concept of race within this framework. The theoretical foundation of Lusitanian Integralism was rooted in Portuguese distinctiveness, idealizing its recreation through a return to ancestral tradition while simultaneously aspiring to imperial greatness.

The invocation of race is deemed indispensable to Lusitanian Integralism, embodying a form of nationalism rooted in tradition. This perspective defines Portugueseness through an ethnic lens (da Cruz 1982) and manifests in the disdain for "internal foreigners" (Martins 2006). Antonio Sardinha explicitly expounded on "the ethnic homogeneity upon which our historical existence is built" (Pinto 1982, p. 1418). He, for instance, characterized the Republic as an "ethnic resurgence of the black people and the Jews" (Ramos 1994, p. 543), employing the attribution of ethnicities to Republican leaders as a means of delegitimizing them.

While the integralist management of the concept of race was not exclusively biological, it did incorporate this element. The racial doctrine does not emerge as a primary contribution of Lusitanian Integralism to the *Estado Novo*—a period marked by a dictatorship akin to fascist ideology. However, it did involve both biological racial hierarchies and opposition to miscegenation (da Cruz 1982). Over an extended period, speeches by dictator Salazar consistently depicted Africans as inferior races. Integralism even played a significant role in promoting anti-Semitic sentiments in the *Estado Novo* (Martins 2006).

The race-based nationalism of Lusitanian Integralism persisted after the Second World War. This is apparent in the writings of Alfredo Pimenta, a Portuguese author who never disavowed Nazi ideology and advocated for racism as a legitimate measure against miscegenation: "Should the people of Angola or Guinea integrate into Portuguese society in large numbers and, through their interbreeding, pose a potential threat to societal balance, (...) does the government not have an obligation to impede such integration and safeguard the purity of our national ethos?" (Marchi 2009, pp. 60–61). In the post-war nationalist press, integralists also advocated for the racial and biological unity of the Portuguese nation. They fervently pledged to defend "Imperial Unity and Greatness" (Marchi 2009, p. 97).

Antonio Sardinha embarked on a journey to "describe the genealogy of the Lusitanian race and present a historical vision of Portugal that validates its racial nationalism" (Pinto 1982, p. 1418). However, he later expressed support for the miscegenation of three races in Brazil—another former Portuguese colony. Sardinha highlighted the "Lusitanian customs in interaction with the land and the natives," ideas that can be recognized as early manifes-

tations of the Luso-Tropicalism theory formulated by Gilberto Freyre, with whom Sardinha maintained correspondence (Gomes 2016, p. 107).

Until the 1950s, the imperial mystique and civilizing mission championed by the *Estado Novo* were rooted in social Darwinism and racial superiority (Castelo 2017). However, starting from that decade, the regime strategically embraced Luso-Tropicalism, largely in response to international pressures favoring decolonization. Therefore, the regime also began promoting a retrospective narrative portraying Portuguese expansion as driven by "the equality of all human beings, regardless of color, race, or civilization" (Castelo 2017, p. 220). There was a significant emphasis on the need to establish multiracial societies in the colonies, especially following the outbreak of war in certain overseas territories.

Supporters of Luso-Tropicalism underscored racial tolerance as a pivotal aspect of Portuguese colonialism (Ribeiro 2004). They propagated this point of view in public opinion, defending the ideals of an interracial society characterized by cultural fusion and without prejudice. The Salazar regime actively erased signs of racial distinction through educational channels, state propaganda, and press censorship. During the 1960s, there was stringent oversight of the use of racializing adjectives in the media. Simultaneously, the advocacy for multiracialism received visual reinforcement in popular culture, particularly through the portrayal of individuals with African phenotypes (Cardão 2012).

After the repeal of the legal-racial hierarchy that persisted in the colonies until 1961, the Portuguese quickly embraced a Luso-Tropical vision (Leónard 1999). This adoption led to the widespread acceptance of a "non-colonial representation" of the Portuguese empire (Cardina 2023). Salazarist nationalism, characterized by a robust emphasis on promoting multiracial harmony, displayed a keen interest in matters pertaining to race and the corresponding discourses. Since 1974, the gradual resurgence of the far-right has been rooted in the mystical legacy of the empire and their stance on the African question (Marchi 2016). Nevertheless, its minor political factions, social movements, and media channels continue to exist on the fringes of Portuguese society.

Marchi (2016, pp. 235–37) highlights a shift away from the Euro-African inclinations ingrained in traditional Portuguese nationalism, a change that unfolded from 1980 onward. This transformation gave rise to an ethno-nationalist discourse, primarily facilitated by the National Action Movement (MAN). The MAN became a hub for a fresh wave of nationalist activists, some of whom were associated with the skinhead subculture. This movement continues to influence the political landscape post-2000, particularly within the National Renewal Party (PNR).

In the realm of the PNR's influence, several anti-immigration organizations, inspired by the European far-right, appeared. One such example is the *Associação Portugueses Primeiro* (APP), established in 2015, which presents itself as committed to cultural struggle. On the other side, the nationalist fanpage *Nova Portugalidade* (NP), established in 2016, aims to rebuild the Lusophone world. With ties to academia, its goal is to promote closer relationships with former Portuguese colonies (Marchi 2023).

The internet's popularity has enabled movements such as APP and NP to use online communication to spread their nationalist ideas (Castells 2009). The web and emerging digital technologies have served as material foundations for the political endeavors of nationalist groups across various countries over time (Berti and Loner 2023; Fuchs 2020; Davis and Straubhaar 2020). This is evident, for example, in far-right movements such as skinheads and Neo-Nazis (Duffy 2003).

In the early stages of web expansion, predictions suggested that this technology might fragment the concept of a nation (Eriksen 2007). However, the reality is that it has enabled various forms of nationalism. Presently, social media platforms serve as political spaces, linking individuals within a network of influence and opposition rooted in identity (Di Fátima 2019; Castells 2009). These digital networks not only reinforce pre-existing nationalist ties established through in-person interactions but also foster new connections achievable solely through online communication (Lajosi and Nyíri 2022).

Previous studies have examined the ideology of nationalist movements in Portugal following the Carnation Revolution in 1974, primarily focusing on the analysis of political programs or militant media (Marchi 2023, 2016, 2010). The exploration of these movements' approach on social media platforms remains very limited; however, it presents several advantages.

On the one hand, posts on these digital platforms make it possible to go beyond abstract ideology and observe discursive actions in a dynamic way. Through the analysis of everyday situations, these posts unveil the practical application of nationalist ideas. On the other hand, the behavior of the movements' followers signifies their engagement with the cause, shedding light on sensitive topics. Through the examination of online behaviors, including actions such as likes, shares, or comments, it becomes possible to identify the enthusiasm that nationalist proposals generate among activists.

### 3. Materials and Methods

This paper explores the processes of racialization within Portuguese society through an analysis of the activities of two prominent nationalist Facebook pages: *Nova Portugalidade* (NP) and *Associação Portugueses Primeiro* (APP). The objective is to answer the following research question: what are the frequency and characteristics of racial allusions found within Portuguese nationalist movements on Facebook? The research employs a mixed-methods approach that integrates both quantitative and qualitative techniques, structured across three procedural stages: data extraction, mining, and visualization. These stages are executed within the framework of Big Data paradigms (Boyd and Crawford 2012).

The data were collected utilizing *CrowdTangle*, a tool developed by Meta, and configured within the Intelligence and Search modules (Fan 2023). This approach facilitates access to official data through the social media Application Programming Interface (API) using the account ID. Subsequently, the data were organized and stored in separate CSV files for each fanpage, capturing both quantitative and qualitative information.

The sample covers 72 months, from January 2017 to December 2020, comprising a total of 3670 posts. This period was selected to standardize the varying dates of fanpage creation and content production. The dataset is highly diverse, encompassing details such as the day and time of the posts, URL addresses, content categories, message text, the number and type of reactions from followers (including likes, shares, love, anger, sadness, haha, and wow), etc. These social data facilitate the analysis of both fanpage administrators' and followers' behavior, grounded in the examination of interaction and engagement rates with the posts (Rogers 2013).

The specialized literature facilitated the compilation of a list featuring 20 Portuguese words commonly employed in processes of racialization within the country (Di Fátima 2023; Miranda et al. 2022; Bartlett et al. 2014). Translating these words proves challenging, as they often lose their intended meaning when rendered into another language. Another challenge is gender. In contrast to English, Portuguese distinguishes between masculine and feminine words, even when referring to a person's nationality or ethnic origin. This list is organized into three analytical categories:

- Geographic origin: africa, africana, africanas, africano, africanos, afro,
- Social identity: negra, negras, negro, negros, negrita, negrito, negritos,
- Skin color: preta, pretas, preto, pretos, de cor, pretalhada, pretalhadas.

The databases underwent processing using *Voyant Tools*, v. 2.6.10, to ascertain the frequency of terms in the text (Baptista et al. 2023) and their vocabulary density (Zaidi and Allahdad 2023). *Voyant Tools* is an open-access software developed in the academic context (Sinclair and Rockwell 2023), used in many fields for textual analysis (Alhudithi 2021), especially for exploring social media posts (Baptista et al. 2023; Schumann 2022; Mihálik et al. 2022; Kiselev 2021).

After that, employing a Visual Basic algorithm, the textual variables were cross-referenced with this designated list of words. The algorithm filters out all posts that contain at least one term from the list, thereby forming a sample of messages characterized by

explicit racial framing. Subsequently, these messages facilitated the creation of a sub-sample consisting of the top posts with the highest number of interactions—16 from each fanpage. This sub-sample was subjected to a framing analysis in two stages:

In the first stage, the research team selected all message parts that included mentions of racial identity. These parts helped to examine how race is discussed (Dixon et al. 2018). The purpose was to identify who is being referred to through racialization and in what thematic contexts. The analysis is supported by tools for discourse analysis applicable to representing social actors (Van Dijk 1997; Van Leeuwen 1997). Specifically, it includes (i) understanding the roles of the individuals mentioned in the post, (ii) observing how individuals and groups are assimilated or collected, and (iii) noting the association or dissociation in defining boundaries between groups.

In the second stage, the research team categorized the frames within the sub-sample. The purpose was to comprehend how fanpages depict racialization across various themes, including immigration, politics, empire, or crime. In this context, the semantic entities under examination could encompass actors, places, events, concepts, etc. (Baden 2018). As the sub-sample comprises lengthy texts, some with multiple frames, the analysis concentrates on the primary frame of each post. Additionally, it assesses whether the frames are episodic or thematic (Iyengar 2005), connecting solely to a specific event or to structural processes.

Studies focusing on frames applied to specific racial categories are more prevalent than research aimed at identifying racializing or ethicizing frames (Lawlor and Tolley 2017; Fox et al. 2012). While racialized frames are occasionally acknowledged, the predominant approach centers on how social actors of different races are portrayed (Dixon et al. 2018; Cranmer et al. 2014). In most of these approaches, it is often implicit that race itself constitutes a form of framing and representation. Racialization, in essence, is a framing of both people and situations. This paper demonstrates that the invocation of race in nationalist discourses encompasses a multi-dimensional framing process: it delineates the social through a frame (Gitlin 2003), defines the framed situation as racialized (Goffman 1986), and imparts specific angles of meaning through persistent enunciative patterns (Entman 1993).

Ethical precautions were also considered. The data have been extracted and processed in an aggregated form, aligning with the recommendations of the *General Data Protection Regulation* (2016/679) currently enforced in the EU. The data are also not publicly available due to privacy concerns. The databases are stored in a secure Google Drive folder, with access limited to individuals responsible for conducting this research. If required, the dataset may undergo an irreversible anonymization process to ensure the safeguarding of sensitive information. The study did not involve the massive collection of personal data, making it impossible to correlate users' profiles on social media with their online behavior (likes, shares, comments, etc.). However, when necessarypeople's faces have been blurred to safeguard their images.

## 4. Results and Discussion

### 4.1. Quantitative Analysis

The fanpages under analysis published 3670 posts on Facebook from January 2017 to December 2020. In this nationalist universe, *Nova Portugalidade* (NP) accounted for 75.5% of the content (n = 2772), while *Associação Portugueses Primeiro* (APP) contributed 24.5% (n = 898). Figure 1 illustrates the distribution of the sample over time, highlighting the power productivity of NP across all years.

The fanpages have similar trends of expansion and retraction in the content creation process. The year 2019, for instance, stands out as an outlier in the sample, accumulating the highest gross number of messages in both cases: NP (n = 625 | 22.5%) and APP (n = 621 | 69.2%). On the other hand, the Covid-19 pandemic, responsible for reinforcing the digitalization dynamics of life in society, does not appear to have accelerated the workflow of fanpages.

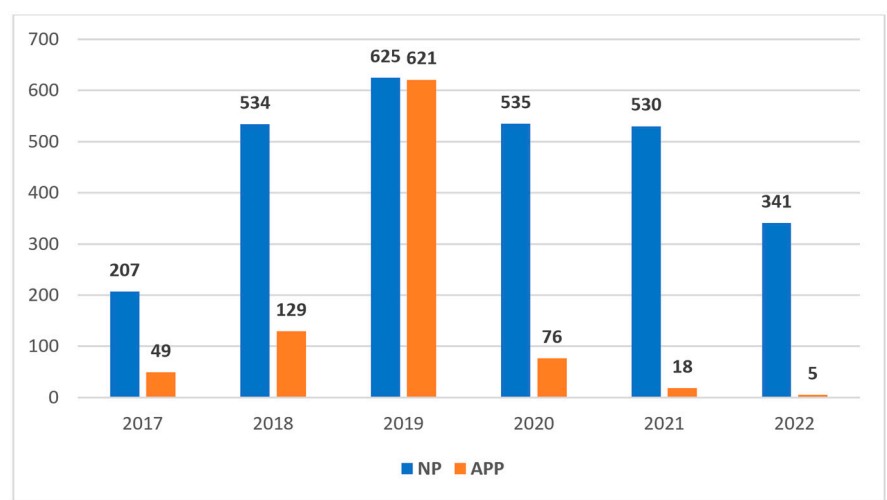

**Figure 1.** Fanpage posts between 2017 and 2022 (N = 3670).

Contrary to the global trend, the years 2020, 2021, and 2022 show a progressive slowdown in production, decreasing from 611 publications to 346 in total—a drop of 43.4%. It is noteworthy that the far-right political party *Chega* was founded in 2019, influencing both online and offline dynamics of Portuguese nationalism. Table 1 highlights the most frequent types of messages in the sample.

**Table 1.** Fanpage post formats (= | %).

|  | Nova Portugalidade | | Associação Portugueses Primeiro | |
|---|---|---|---|---|
|  | **n =** | **%** | **n =** | **%** |
| Image | 2555 | 92.2 | 219 | 24.4 |
| Link | 141 | 5.1 | 524 | 58.4 |
| Video | 75 | 2.7 | 138 | 15.3 |
| Status | 1 | 0.0 | 17 | 1.9 |
| Total | 2772 | 100.0 | 898 | 100.0 |

The fanpages employ different strategies on Facebook. Static images, including photos, cartoons, and memes, constitute 92.2% of the content on NP. This is followed, albeit at a significant distance, by links (5.1%), videos (2.7%), and status updates (0.0%). Commonly featured images encompass national symbols, portraits of historical figures, and reproductions of documents, all contributing to a frequent revival of pride in the homeland's significant achievements.

The concept of Portugal as a diverse and inclusive nation is consistently portrayed in the content. Notably, there is a recurring discourse associated with promoting the *Estado Novo*. The content also extends a welcoming embrace to individuals from former colonial African countries and their Portuguese descendants. These individuals are embraced, as they symbolize a mythical moral superiority from the Portuguese empire, emphasizing Lusophone multiculturalism.

On the other hand, the APP exhibits a better balance among various post formats. Links occupy a substantial portion of the content (58.4%), followed by images (24.4%), videos (15.3%), and status updates (1.9%). These URLs direct followers to various websites, primarily focusing on news from national media outlets. The *Observador* website leads the ranking, comprising 14.3% of the linked messages, followed by *Notícias Viriato* (7.1%), *Diário de Notícias* (6.5%), *Sol* (6.1%), *Público* (5.0%), *Expresso* (4.8%), and *Correio da Manhã* (4.4%). Notably, the news content is employed for anti-immigration rhetoric, portraying individuals from former African colonies and their Portuguese descendants as social scum.

Figure 2 shows the growth of fanpages analyzed over the years. Productive activity may have driven the increase in followers, although this growth has not always been consistent. The data also reveal periods of slight contraction. NP recorded a total growth of 129.9%, with 2020 standing out as the year with the most significant productive flow, showing a 33.3% increase in followers. Once again, the COVID-19 pandemic does not seem to have had a significant impact. This phenomenon is also evident in the APP. With a total growth of 320.6%, the most significant increases were observed in 2017 (132.5%) and 2018 (135.2%), respectively, thanks to the expansion of the fanpage in its early years.

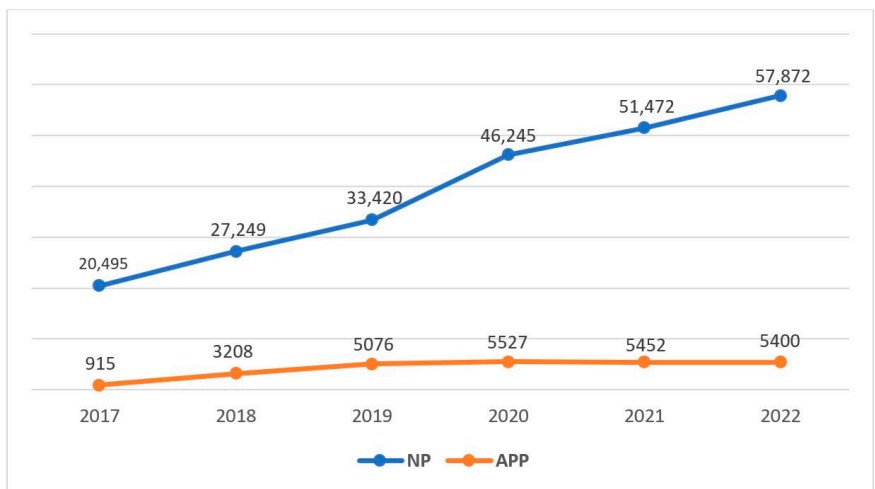

**Figure 2.** Fanpage followers between 2017 and 2022 (N = 262,331).

The APP fanpage also experienced contraction during the period of this research, specifically in 2021 (−1.37%) and 2022 (−0.96%). This outcome might suggest a low adherence of Portuguese followers to the extreme nationalist discourse on social media. However, this phenomenon is not necessarily corroborated by the way followers engage with the posts. This virtual relationship is established through interaction with the content (likes, shares, comments, etc.), though it is important to note that not all actions carry the same weight or significance. These interactions also serve as a means of signaling the convictions advocated by users to their distant connections, including friends, colleagues, and family members, via the profile's news feed. In many cases, these interactions can reflect the deepest anxieties and desires of the community.

The fanpages garnered just over 1.4 million interactions on Facebook between 2017 and 2020 (NP = 88.1% | APP = 11.9%). Upon analyzing the individual categories, the most significant findings underscore the necessity for APP followers to share and thereby amplify the reach of anti-immigration messages. While 22.4% of NP interactions constitute shares, these data escalate to 42.4% on APP—nearly double. In this digital ecosystem, the act of sharing reinforces a cosmology rooted in identity. It is as if to say: *This is who I am, and I am so proud of it that I even share it with others*. Table 2 reveals another facet of this digital scenario, marked by the proliferation of online hate speech.

Emotional reactions represent around 12.5% of the sample's interactions. However, the fanpages have the opposite results. Posts by NP, based on homeland and past imperial pride, have more Love over time (69.9%). On the other hand, content by APP, focused on anti-immigration, mobilizes more Angry (60.9%). The point of convergence is the Sad reaction. The fanpages are almost evenly matched, at 12.0% each. The opposition of these reactions and, at the same time, their strange proximity can be summarized in a maxim: *One's heaven can be another's hell*. Figure 3a,b shows the posts with the highest Love and Angry reactions, expressing this ambiguous conflict of affection and repulsion. People's faces have been blurred to safeguard their personal data and images.

**Table 2.** Emotional reactions from Facebook followers (N = ǀ %).

|  | Nova Portugalidade | | Associação Portugueses Primeiro | |
|---|---|---|---|---|
|  | **n =** | **%** | **n =** | **%** |
| Love | 93,063 | 69.9 | 4144 | 9.6 |
| Sad | 16,067 | 12.1 | 5350 | 12.3 |
| Angry | 11,249 | 8.4 | 26,402 | 60.9 |
| Care | 3092 | 2.3 | 9 | 0.0 |
| Wow | 4368 | 3.3 | 1841 | 4.2 |
| Haha | 5339 | 4.0 | 5625 | 13.0 |
| Total | 133,178 | 100.0 | 43,371 | 100.0 |

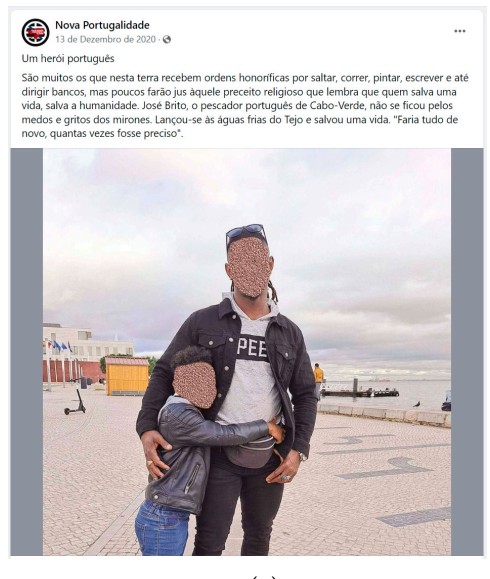

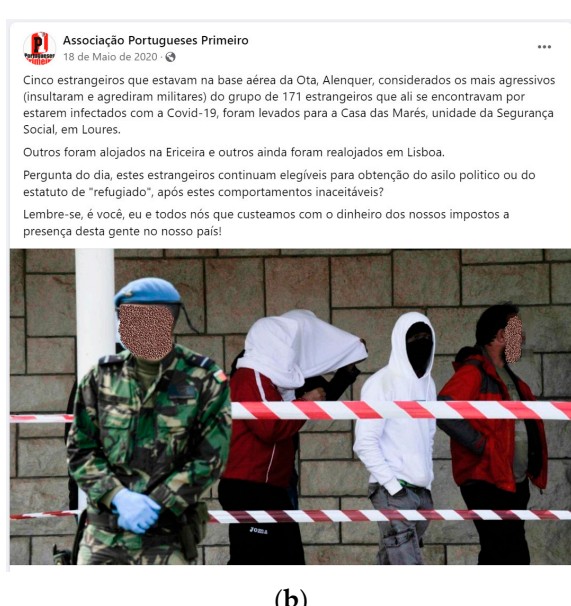

| (**a**) | (**b**) |
|---|---|

**Figure 3.** Posts with the most Love and Angry on the fanpages. (**a**) It is the message with the most Love reactions on *NP* (n = 2404). (**b**) It is the message with the most Angry reactions on *APP* (n = 353).

Published on 18 May 2020, the post that sparked the most anger among APP followers featured a photo of a supposed air base where immigrants had clashed with military personnel in a pandemic context. Its text, infused with irony, questioned: "Are these foreigners still eligible for political asylum [sic] or 'refugee' status after these unacceptable behaviors?" The post then issued a warning: "Remember, it's you, me, and all of us who are footing the bill for these individuals' presence in our country with our tax money!" (see Figure 3b).

A few months later, on 13 December 2020, the post that garnered the most love from NP's followers featured a photo of two young black men, praising an act of courage. The accompanying text narrates the touching feat: "A Portuguese fisherman from Cape Verde did not heed the fears and cries of onlookers. He plunged into the cold waters of the Tagus River and saved a life." (see Figure 3a).

The two posts focus on the same mythical character. However, they present opposite stories. In the first post, the immigrant is described as a social parasite, a troublemaker, living off the state. Conversely, in the second, the immigrant is portrayed as a national hero, an upholder of order, willing to risk his life for his neighbor. At least in this regard, love triumphed over hate. In the weighted calculation, the message infused with more Love engaged 52.1% of the fanpage's followers, while Angry mobilized 29.5%. This affective

opposition is confirmed by the most frequent words in the sample. Figure 4a,b shows this narrative difference through word clouds.

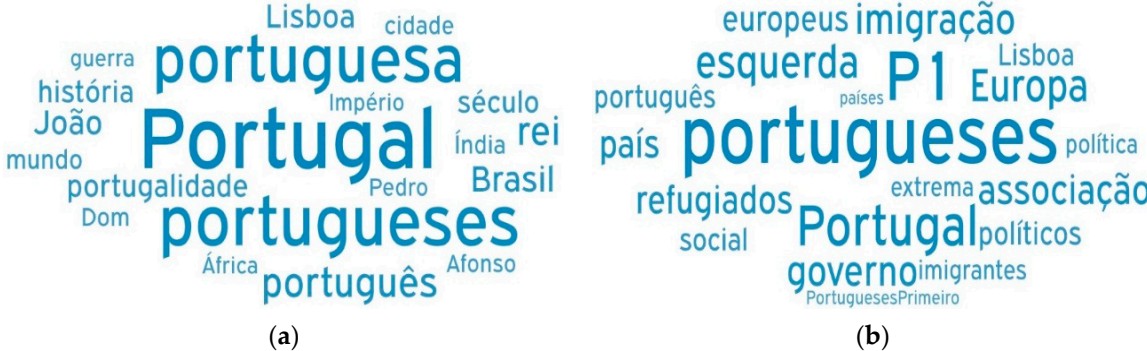

(**a**)                                                                          (**b**)

**Figure 4.** Most frequent Portuguese words on fanpages (N = 20). (**a**) Words used on *NP*. (**b**) Words used on *APP*.

The fanpages published 3670 posts from January 2017 to December 2020. Approximately 97.5% of them have a textual description (N = 3581), with higher rates on NP (99.1%) than on APP (92.8%). Despite the strong multimedia component of the content, the data underscore the significance of the written word on social media. On average, the NP fanpage uses 35.0% more words in sentences than APP (25.9 versus 19.3), possibly attributable to the meticulous descriptions of historical events. Conversely, APP exhibits a higher vocabulary density (0.138 versus 0.056), suggesting the presence of potentially more diverse and complex texts to read. It is also plausible that the repetition of certain words in the description of historical events affected the NP results, or that APP simply deals with a broader range of themes.

The ten most frequent words on NP are Portugal (3819), portugueses (2558), portuguesa (2303), português (2043), rei (1936), Brasil (1532), João (1354), Lisboa (1271), portugalidade (1168), and história (1111). The use of these terms is geared towards praising famous deeds and their characters over the centuries, both in Portugal and in the former colonies (see Figure 4a). The image of a small but particularly remarkable country is constructed.

On the other hand, the ten most frequent words on APP are portugueses (599), P1 (450), Portugal (302), esquerda (203), Europa (199), imigração (175), governo (171), associação (160), refugiados (148), and país (147). These terms are used to directly attack immigrants, left-wing parties, and the political actions of the socialist government. The image of foreigners is constructed as a national and European threat, especially those from the former colonies (see Figure 4b).

These results confirm the dichotomy of the fanpages, wherein narrative elements converge and diverge. Despite encountering the same words in numerous texts—there is even an exact match of core terms such as Portugal, Portuguese, and Lisbon—their contextual application varies based on the analysis of social phenomena. To delve into racial frameworks, it is essential to utilize a sub-sample generated from a list of racialization words. Table 3 displays this sub-sample categorized (n = 818).

**Table 3.** Posts with explicit racial framework (N = | %).

|  | Nova Portugalidade | | Associação Portugueses Primeiro | |
|---|---|---|---|---|
|  | **n =** | **%** | **n =** | **%** |
| Geographic origin | 581 | 21.1 | 48 | 5.8 |
| Social identity | 230 | 8.4 | 13 | 1.6 |
| Skin color | 183 | 6.7 | 15 | 1.8 |
| Total | 750 | 27.3 | 68 | 8.2 |

The results indicate that 35.6% of the posts contain at least one term related to racialization. This usage is proportionally more common in NP publications (27.3%) than in APP (8.2%). On the other hand, the category that pertains to geographic origin takes the lead on both fanpages, representing 26.9% of the total, followed by social identity (10.0%) and skin color (8.5%). It is undeniable that the former African colonies and their peoples constitute a significant target of racial framing, frequently experiencing direct attacks of hatred. From this point, a qualitative analysis was conducted on the posts with the highest engagement on Facebook.

### 4.2. Qualitative Analysis

The initial stage of qualitative analysis involves examining the main themes. This step reveals the specific areas where the two nationalist fanpages consider race to be significant. When these fanpages incorporate racial categories or employ racially charged language in their discussions, they frame events or individuals in terms of race rather than considering alternative perspectives. Table 4 shows how the two fanpages employ racial framing.

**Table 4.** Racial framing of posts with the most interactions (N = 32).

| Nova Portugalidade | | Associação Portugueses Primeiro | |
|---|---|---|---|
| **Themes** | **Posts = 16** | **Themes** | **Posts = 16** |
| Portuguese empire | 4 | Immigration | 7 |
| Colonial war | 3 | Politics | 3 |
| Racism | 3 | Racism | 2 |
| Politics | 2 | Crime | 2 |
| Decolonization | 2 | Slavery | 1 |
| Former colonies | 1 | Religion | 1 |
| Sport | 1 | - | - |

The nationalism expressed by the APP strongly associates race with the present. Its posts have the most significant impact in two specific discourses: (i) when it portrays race as a key factor in situations deemed unfavorable, such as immigration and crime; and (ii) when it attempts to reject, reverse, or criticize ideas related to racial minorities, such as multicultural policies and quotas. In contrast, NP nationalism leans towards recalling historical events, considering the present except when current situations prompt reflections on the past.

In essence, APP is influenced by a form of nationalism that imparts a racial perspective to contemporary politics, consistently portraying African and black people in an overwhelmingly negative light. The fanpage disregards individuals from these communities in national events unless it contributes to their stigmatization. On the other hand, NP does not identify significant racial elements in present day. While the fanpage seldom discusses the current context in its posts, it interprets the historical narrative through a racial perspective, spanning from the 16th century to 1974.

Analyzing how social actors are depicted in relation to association/dissociation, genericization, or collectivization, fanpages display clear discursive features in how they use the terms African and black people. In forming their ideas about Portuguese identity, one fanpage is distinctive for excluding blackness, while the other incorporates it. Notably, the APP consistently refrains from connecting the identities of African and black people with Portuguese identities. The racial category serves as a criterion for determining who is included or excluded from the national identity, based on a biologically rooted racist ideology.

When APP acknowledges the presence of "Afro-descendants who have acquired Portuguese citizenship", it does so solely to link them to criminal activities. Following this, the fanpage raises the question: "What percentage of individuals with African heritage are

actually convicted in court?" These posts initiate a semantic shift, aligning descendants of immigrants with a racial categorization—an aspect emphasized and considered unchangeable by this ethno-nationalistic perspective. This discourse on racialization is part of a broader pan-European ethno-nationalism, evident in discriminatory slogans akin to those used by other movements and reproduced by APP: "if you are black, go back", as seen in the *Conservative People's Party of Estonia* (ERKE).

On the other hand, NP mobilizes African and black people, aligning them perfectly with the Portuguese national category. Individuals and groups labeled as black are consistently also identified as Portuguese. This occurs across various situations and historical periods, but primarily within the context of empire. The fanpage consistently blends nationality and race in a meticulously harmonious manner: from a 16th-century samurai described as "black, Mozambican, and Portuguese" (a notable convergence) to the identification of the "first black Portuguese parliament member" (in 1965, in the National Assembly of the *Estado Novo*). The APP's narrative also encompasses "mestizaje", a term forbidden in the Portuguese ethno-nationalist vocabulary.

When employing racial categories, one fanpage predominantly uses the term *African(s)*, while the other prefers the term *black(s)*. APP's racially exclusionary nationalism favors using the term *African(s)*. The intention is to euphemize biological racism and present it as a rejection of cultural differences. In contrast, the NP's inclusivist nationalism frequently employs the explicitly racial term *black(s)*. This serves as a clear means of emphasizing its commitment to multiraciality.

The prevalence of generic and collective representations of Africans in the APP reveals its perspective on a world characterized by racial classes. The individuals depicted are viewed as exemplars. Conversely, NP's discourse more frequently emphasizes individualization and the naming of racialized subjects, typically conveying a positive tone. The following posts show these aspects:

- APP (8 January 2020): "The faces of the individuals responsible for the young man's death in Campo Grande have been disclosed. They are all Africans from Guinea, and this is their way of expressing gratitude for their time in Portugal (. . .)".
- NP (7 April 2018): "Black, Mozambican, and Portuguese: the remarkable life of Yasuke, the Mozambican samurai who will be the focus of a movie. Being black, Mozambican, and Portuguese is natural; being black, Portuguese, and a samurai is less common (. . .)".

Racial categories serve as primary definitions. Consequently, references to race are embedded in broader discourses with specific perspectives and meanings. Race functions as both a frame and an object of framing. Beyond the racializing segments, it is possible to capture frames. This allows for differentiation in how each nationalist fanpage articulates its conception of race within broader themes. One general characteristic of the sub-sample is the tendency for posts to avoid using episodic frames and instead employ thematic ones. The narrative portrays specific cases as long-term structural phenomena.

As a distinct form of nationalism, NP is guided by two frames: (a) multiracialism and (b) defense of the nation. A notable feature of the former is its ability to span across various subjects. This indicates that diverse subjects can be connected through shared structures that guide their definition in a unified direction.

The most expressive frame is the multiracialist perspective, encompassing various themes such as empire, politics, and sport. These perspectives emphasize the sense of belonging and loyalty of black people to Portugal. For example: the military officer Marcelino da Mata (who fought for the Portuguese army against the independence movement in Guinea) and the athlete Patrícia Mamona (who asserts her Portugueseness regardless of race). The common thread among all these cases is their alignment with the vision of *Estado Novo* and the Luso-Tropicalism ideology in shaping the nation. A similar perspective is adopted for the war in African territories during the dictatorship. By applying the concept of "overseas war", many posts employ the framework of national defense and manipulate racial elements to justify this historical event.

The coexistence of racially distinct individuals is portrayed in APP discourse as opposition, threat, or incompatibility. Examples of these frames include (a) insecurity and ethnic conflict, (b) national purity, and (c) cultural stigmatization. Within any of these frames, racial categories are employed to underscore disparities or heighten conflicts: whites, Portuguese, and Europeans *versus* blacks, immigrant, and Africans. Table 5 illustrates the cross-referencing relationship between framing and themes.

**Table 5.** Exclusive frames and themes from each nationalist fanpage.

|  | **Associação Portugueses Primeiro** | | | **Nova Portugalidade** | |
|---|---|---|---|---|---|
| **Frames** | Insecurity ethnic conflict | National purity | Cultural stigmatization | Multiracialism | National defense |
| **Themes** | Immigration Crime | Immigration | Religion | Empire Sport Colonial war Politics | Colonial war |

APP's posts on immigration are presented as instances of insecurity, ethnic conflict, or national purity. They convey the notion of invasion and demographic replacement. These narratives often adhere to a specific storytelling structure. They introduce a problem (immigrants, minorities), proceed to assign blame (governments, left-wing parties, NGOs), and ultimately propose a solution (ethnic exclusion). On the other hand, NP's posts center on national defense and multiracialism, integrating historical figures from former Portuguese African colonies.

While the fanpages exhibit opposing perspectives on blackness, the sub-sample reveals similarities in their discourses. Table 6 shows the shared frames, highlighting the denial of post-colonial reparations as the most prevalent theme. Both nationalist discourses approach the issue as a point of contention, aiming to contest, delegitimize, or reverse claims acknowledging the damage caused by colonizing countries in Africa.

**Table 6.** Frames shared by both fanpages.

| **Frames** | **Denial of Post-Colonial Reparations** | | **Refuting (anti-)Racism** | |
|---|---|---|---|---|
| **Fanpage** | **APP** | **NP** | **APP** | **NP** |
| **Themes** | Immigration Slavery | Empire Racism Politics Decolonization Ex-colonies | Politics Racism | Empire Racism |

Two posts effectively illustrate the alignment of the nationalist fanpages:

- APP (3 March 2020): "The Africans, who enslaved their black brothers for so long (and still do today in some areas), certainly don't recall constructing a similar mausoleum. Why should Europeans do it? Stop scrutinizing the past through the lens of the present and refrain from inciting hatred."
- NP (29 May 2022): "If there are still accounts to be settled in the sluggish bookkeeping of the feeble-minded, here is an instance of Portuguese retribution in Africa. Benguela School of Arts and Crafts in 1965."
- The first post addresses the proposed construction of a monument dedicated to slavery, sponsored by the Lisbon City Council. In contrast, the second post depicts young people with diverse phenotypes in a classroom—a visual theme extensively employed in Luso-Tropicalism propaganda. In both cases, there is framing that relativizes the role of the colonizers or presents it as beneficial.
- This approach persists across various themes. Nonetheless, the NP also uniquely frames some speeches within a logic of reparation—not for any debts owed by Portugal

to the colonized, but rather to the former colonizers. "The Portuguese African people" are depicted as creditors seeking an "apology from the Portuguese state" in response to decolonization and its circumstances.

- Refuting racism and anti-racism are another shared frame. However, the fanpages do this in different ways. On one hand, APP refutes accusations of racial discrimination against minorities by presenting supposed instances of favoritism. For example: residing in unfinished buildings in a poor neighborhood without paying for water or electricity would not be permitted for "the rest of the citizens". On the other hand, NP advocates for equality between black and white people. For example: "the majority of the white Portuguese was also poor and of humble status [ in the former colonies], just as hardworking as the black Portuguese majority". Therefore, racism exists against the majority in the ethno-nationalist framework, whereas racism was never present in the context of Luso-Tropicalism, much the same as colonialism.

Anti-racism faces different forms of delegitimization. One example is the rejection of the *Black Lives Matter* movement. NP's Luso-Tropicalism undermines this movement by framing it as an out-of-place slogan in Portugal—a nation that "ordained the first black bishops" and "elevated hundreds of blacks to the status of royalty". Conversely, APP's ethno-nationalism attempts to counter the charge of racism by alleging that the movement's activists do not assign equal value to "white lives". In the end, the frames that include both types of nationalism show that the apparent deep disagreement between the APP and the NP about racial compatibility and national belonging converge in meaning when addressing contemporary issues.

## 5. Conclusions

This paper explored the process of racialization in Portuguese right-wing political movements, focusing on two prominent nationalist fanpages: *Nova Portugalidade* (NP) and *Associação Portugueses Primeiro* (APP). The objective was to answer the following research question: what are the frequency and characteristics of racial allusions found within Portuguese nationalist movements on Facebook? Data were collected using *CrowdTangle* covering 3670 posts spanning 72 months, from January 2017 to December 2020. The analysis focuses on the online behavior of fanpage administrators and their followers, examining interaction and engagement.

Nationalist movements in Portugal have experienced significant growth on social media in recent years, garnering an increasing number of followers (see Figure 2). Impressively, these two established fanpages alone generated 1.4 million interactions on Facebook. Future studies should consider incorporating smaller and peripheral fanpages to accurately assess the true impact of these online movements.

Emotional responses played a significant role in the interactions, albeit in opposite directions for each fanpage (see Table 3). While NP's posts, based on the homeland and imperial pride of the past, evoke a greater sense of Love (69.9%), APP's anti-immigration content tends to stir more Anger (60.9%). Other results confirm this dichotomy. For example, fanpages use identical words in many texts but create practically opposite meanings with them. Just over a third of the posts contain at least one term related to racialization, with significantly varied occurrences across fanpages: NP (27.3%) and APP (8.2%).

While assertions have been made regarding the replacement, within the far-right, of Portuguese Euro-African nationalism by an ethno-nationalist inclination aligned with European xenophobic movements (Marchi 2016), the main results of this research indicate that Portuguese nationalism maintains online a dual ideology concerning race. This observation holds true irrespective of whether fanpage are categorized as far-right or non-radical. These dual nationalisms are of particular concern to individuals from Africa and African descent (see Figure 3). APP rejects miscegenation and equates national identity with whiteness. NP asserts the Portuguese capacity to assimilate different races under a national umbrella without racism.

Luso-Tropicalism propels the nationalist fanpage NP, amassing a greater number of followers and global interactions on Facebook compared to APP's ethno-nationalism. After a gap of ethno-nationalist ideas in the public space for some decades, the prosperous far-right political party *Chega* has the potential to instigate shifts within the nationalist movement. On one hand, the resurgent nationalist movement often amalgamates these two trends. On the other hand, social media is the base of its organization, both to recruit followers and to promote ideas.

Race is revealing itself as a somewhat arbitrary and opportunistic concept within the Portuguese nationalist movement. Various stances can be discerned not only among factions based on their ideological structures but also within each fanpage. Depending on convenience, the discourse surrounding race may highlight or downplay it. Moreover, it might be perceived as an irreducible identity or completely disregarded.

There is a clear strategy of racialization or de-racialization in these movements on Facebook, particularly on the Luso-Tropicalist fanpage. Their framing implies a specific meaning: while it may be reasonable to racialize people by referencing the past (to defend a false absence of discrimination), the use of racial labels should be avoided in contemporary Portuguese society.

NP's Luso-Tropicalist nationalism uses a racialist narrative to deny racism, manipulating historical contexts while emerging as one of the rare political movements openly employing such labels within the Portuguese public sphere. On the other hand, it is paradoxical that an evidently racist movement such as APP (in the sense that it uses race as a criterion for exclusion) uses racial labels not so often. This is probably due to linguistic reasons, and future studies could explore it. Even ethno-nationalist activists realize the stigma of so grossly offending the anti-racist norm.

The identified types of nationalism also delineate two paths of symbolic competition, wherein the Portuguese find themselves contending with various groups. In the context of APP ethno-exclusivism, white and European Portuguese directly vie with immigrants and other races for the exclusive legitimacy of representing a racialized nation. In the case of NP ethno-pluralism, the implicit symbolic competition extends to other powers and former colonizers. There is an attempt to establish moral superiority rooted in mythical narratives detailing how the Portuguese interacted with former colonized peoples. While Portuguese nationalism's perspective on race incorporates both an ethno-exclusivist and an ethno-pluralist narrative, it would be inaccurate to label the second one as strictly anti-racist. Both NP and APP strongly oppose anti-racist movements on the internet. Despite their disparities, they can be pragmatic allies in realpolitik.

This research demonstrated that the invocation of race in nationalist discourses encompasses a multi-dimensional framing process. In this sense, analyzing empirical data from social media offers two advantages. Firstly, it enables the exploration beyond broad ideologies, allowing for the scrutiny of the discursive actions of movements in a dynamic manner. Through the examination of concrete events, Facebook posts reveal the incremental practical implementation of their ideologies. Secondly, it offers the opportunity to evaluate the level of interaction each post garners and to construct a sub-sample of the most widely echoed content. This facilitates the identification of the types of discourse that genuinely resonate the most with sympathizers of right-wing and far-right nationalist movements.

The main limitation of this research is the focus on analyzing only two pages on Facebook. While APP and NP are considered the most popular on social media, they may inadvertently overlook minority aspects of nationalism. Consequently, it becomes impossible to extrapolate the findings to the entire Portuguese nationalist movement on social media. Some of these movements also maintain activity on other platforms, albeit with significantly fewer posts, operational time, and followers. Future studies could integrate all platforms, thereby expanding the understanding of these nationalist movements on social media.

**Author Contributions:** Conceptualization, J.R.C.; methodology, B.D.F.; formal analysis, B.D.F. and J.R.C.; investigation, B.D.F. and J.R.C.; data curation, B.D.F. and J.R.C.; writing—original draft

preparation, B.D.F. and J.R.C.; writing—review and editing, B.D.F. and J.R.C. All authors have read and agreed to the published version of the manuscript.

**Funding:** This research received no external funding.

**Institutional Review Board Statement:** Not applicable. No ethical approval was required. The research analyzes official data aggregated from Facebook and did not involve human beings or personal data.

**Informed Consent Statement:** Not applicable. The research did not involve humans and not consent was required.

**Data Availability Statement:** The data presented in this study are available upon request from the corresponding author. The data are not publicly available due to privacy concerns, as elucidated in the Data and Methods section of this paper.

**Conflicts of Interest:** The authors declare no conflict of interest.

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
