# Peer review of "One’s Heaven Can Be Another’s Hell: A Mixed Analysis of Portuguese Nationalist Fanpages"

_socsci, doi:10.3390/socsci13010029_

Round 1
Reviewer 1 Report
Comments and Suggestions for Authors
1. The manuscript clearly furthers our understanding on Portuguese nationalist fanpages. As stated, these pages pay a key role in the shaping of extreme right-wing movements. By analyzing Nova Portugalidade and Associação Portugueses Primeiro, the authors identified a dual ideology and different communication strategies as the main findings.
The paper develops a robust methodological approach based on data extraction, mining, and visualization. The use of 20 words commonly employed in processes of racialization within the country makes full sense as they are related with analytical categories. Besides that, Voyant Tools helps to the textual analysis, but it would be recommended to reinforce the validity of this tool by citing recent research based on the same strategy. For instance, see this paper on the Portuguese context:
-Baptista, J. P., Rivas-de-Roca, R., Gradim, A., & Loureiro, M. (2023). The Disinformation Reaction to the Russia–Ukraine War: An Analysis through the Lens of Iberian Fact- Checking. KOME, 11(2). https://doi.org/10.17646/KOME.of.2
2. The article has a scientific structure. Nevertheless, there are no formal objectives or research questions. This information is needed for a better structure of the paper, especially regarding the conclusions. Conversely, tables and figures are quite illustrative.
Maybe it would be helpful to come back to objectives, research questions or even hypotheses in a broader way in the Conclusions. This section is too short. Although the implications of framing are addressed in the discussion, further details should be given to contextualize the results. On this matter, future lines of research or limitations are required.
3. The list of references is up-to-date and suitable for the present study. However, I miss more international studies on the populist far-right. The background section is based on extreme right-wing nationalist movements, but only focused on Portugal. Even though the Portuguese situation is essential to understand the study, the authors may consider other contributions on the worldwide growing of the far-right :
-Mudde, C. (2016). Europe’s populist surge: A long time in the making. Foreign Affairs 95: 25–30.
-Wodak, R. (2015). The Politics of Fear: What Right-Wing Populist Discourses Mean. London: Sage
4. The sample is huge, covering 72 months and 3,670 posts on Facebook. For its part, the analysis is deep as it shows the posts formats, the number of followers, the emotional reactions, or the presence of racial contents. Migration and emotion are central elements for the populist communication style; hence, this paper allows that one can clearly learn a lot about the working of nationalist fanpages in a context (Portugal) where far-right has grown in recent years.
5. Lastly, the discussion is sometimes worded confusingly. I would recommend identifying two or three empirical contributions of this paper to the line of inquiry.
6. In short, this paper is an interesting approach for the political and communication field, particularly aligned with prior scholarship on fanpages and their connection with extreme right-wing movements which resort to mythical concepts such as an idealized past. Only some minor changes are requested as reinforcing the theoretical framework results, presenting clear objectives/research questions, or rebuilding the conclusions.
Author Response
Dear Reviewer,
Thank you for thoroughly reviewing our paper and providing valuable comments. We appreciate your recognition of the relevance of the findings in the field of right-wing and far-right nationalism studies. We have incorporated nearly all your suggestions within the limited timeframe.
1. Voyant Tools is free software developed within the academic domain and extensively employed for textual analysis. A robust theoretical framework exists regarding its application across diverse science fields. We used the reference you recommended along with other citations linked to the utilization of the tool and the analysis of social media.
2. We incorporated the research question into both the introduction and the methods sections, revisiting it in the paper's conclusions. Additionally, the conclusions now offer more detailed contextualization of the results, highlighting potential avenues for future research and acknowledging the primary limitations.
3. Thank you for bringing this to our attention. The concept of "populist far-right" was not explicitly discussed in our paper. The movements under analysis and their activities on Facebook may not precisely fit within this definition, and it could certainly be a topic for extensive debate. We consistently refer to right-wing nationalism, with the term "far-right" specifically applying to APP fanpage.
4. Thanks for the review. We completely agree with you.
5. We have also made specific modifications to improve comprehension. Additionally, a significant part of the conclusions has been reorganized. We present the main quantitative data from the research and draw correlations with qualitative results: fanpages growth, emotional engagement, narrative dichotomy, and ideological stances.
6. Thank you for your valuable comments. We have made every effort to incorporate as many of your suggestions as possible within the limited time frame. If you have any more questions, we are at your disposal.
Reviewer 2 Report
Comments and Suggestions for Authors
I do this kind of research and found this interesting. The study raises an intriguing conceptual question about the uses of the past vs. uses of the present that I wish were explored more. If there is a rewrite, I would like to know the effect sizes of the differences described in Table 2, 3 and 4 (probably a Cramer's V?).
I raised an ethics question that doesn't not affect the research itself and whose resolution I'm leaving to the editor and/or copy editor. There are some direct quotes from the Facebook posts, and as I read the EU General Data Protection Regulations, these should not be used because I could then locate the individual in FB. I'm going to leave this to the editor and/or copy editor because I am sure there are different interpretations of these guidelines. If the direct quotes are not allowed, the alternative would be to paraphrase them.
Author Response
Dear Reviewer,
Thank you for thoroughly reviewing our paper and providing valuable comments. We appreciate your recognition of the relevance of the findings in the field of right-wing and far-right nationalism studies. It is very gratifying to hear this from an expert in our field.
The paper underwent a rewriting process, significantly influencing the methods and conclusions. We tried to address all the referees' suggestions within a limited timeframe. Thank you for your meticulous analysis. While we know Cramér's V measure for the chi-squared test, unfortunately we don't have the necessary knowledge to apply it. We come from the Communication Sciences. We will be paying more attention to this in future studies.
We appreciate the ethical concerns that have been raised. The quoted content originates from public posts on fanpages associated with political movements and has been extracted using an official tool provided by Meta. It is important to note that these quotes are not attributed to individual users and, in most cases, do not involve personal data. Despite this, the original posts were in Portuguese and have been freely translated into English, making tracking them more difficult. In the revised edition of the paper, we have incorporated measures to enhance data protection. In Figure 3, facial features of individuals have been blurred to safeguard their images.
Thanks again for your valuable comments. We have made every effort to incorporate as many of your suggestions as possible within the limited timeframe. If you have any more questions, we are at your disposal.